# Frailty in randomized controlled trials of glucose-lowering therapies for type 2 diabetes: An individual participant data meta-analysis of frailty prevalence, treatment efficacy, and adverse events

**Heather Wightman**[1], **Elaine Butterly**[1], **Lili Wei**[1], **Ryan McChrystal**[1], **Naveed Sattar**[2], **Amanda Adler**[3], **David Phillippo**[4], **Sofia Dias**[5], **Nicky Welton**[4], **Andrew Clegg**[6], **Miles Witham**[7,8], **Kenneth Rockwood**[9], **David A. McAllister**[1], **Peter Hanlon**[1]*

1 School of Health and Wellbeing, University of Glasgow, Glasgow, United Kingdom, 2 School of Cardiovascular and Metabolic Health, University of Glasgow, Glasgow, United Kingdom, 3 Diabetes Trials Unit, University of Oxford, Oxford, United Kingdom, 4 Population Health Sciences, Bristol Medical School, University of Bristol, Bristol, United Kingdom, 5 Centre for Reviews and Dissemination, University of York, York, United Kingdom, 6 Academic Unit for Ageing and Stroke Research, Bradford Teaching Hospitals NHS Foundation Trust, University of Leeds, Bradford, United Kingdom, 7 AGE Research Group, Translational and Clinical Research Institute, Faculty of Medical Sciences, Newcastle University, Leeds, United Kingdom, 8 NIHR Newcastle Biomedical Research Centre, Newcastle upon Tyne NHS Foundation Trust, Cumbria Northumberland Tyne and Wear NHS Foundation Trust and Newcastle University, Newcastle, United Kingdom, 9 Division of Geriatric Medicine, Dalhousie University and Nova Scotia Health, Halifax, Nova Scotia, Canada

* Peter.hanlon@glasgow.ac.uk

## Abstract

### Background

The representation of frailty in type 2 diabetes trials is unclear. This study used individual participant data from trials of newer glucose-lowering therapies to quantify frailty and assess the association between frailty and efficacy and adverse events.

### Methods and findings

We analysed IPD from 34 trials of sodium-glucose cotransporter 2 (SGLT2) inhibitors, glucagon-like peptide-1 (GLP1) receptor agonists, and dipeptidyl peptidase 4 (DDP4) inhibitors. Frailty was quantified using a cumulative deficit frailty index (FI). For each trial, we quantified the distribution of frailty; assessed interactions between frailty and treatment efficacy (HbA1c and major adverse cardiovascular events [MACE], pooled using random-effects network meta-analysis); and associations between frailty and withdrawal, adverse events, and hypoglycaemic episodes. Trial participants numbered 25,208. Mean age across the included trials ranged from 53.8 to 74.2 years. Using a cut-off of FI > 0.2 to indicate frailty, median prevalence was 9.5% (IQR 2.4%–15.4%). Applying a higher threshold of FI > 0.3, median prevalence was 0.5% (IQR 0.1%–1.5%). Prevalence was higher in trials of older people and people with renal impairment however, even in these

**Data availability statement:** This study used individual participant data from industry-sponsored randomized controlled trials. The trial data are available upon application to the trial sponsors via the Vivli analysis platform (https://search.vivli.org/) subject to a data sharing agreement. This data sharing agreement stipulates that all individual-level data are held securely within a remote analysis platform. All model outputs from our analyses, and aggregated data and summarized statistics are included within the supplementary appendix, along with analysis code on which all of the presented results are based (S2 Appendix).

**Funding:** This analysis was funded by a grant from Tenovus Scotland to PH and DM (S22-27: Assessing frailty and representativeness in randomized controlled trials of glucose lowering therapies for type 2 diabetes). The systematic review, identification, and acquisition of IPD (within which this study was nested) was funded by the Medical Research Council to DM, EB, NW, and SD (MR/T017112/1: Routine care treatment effectiveness in people with type 2 diabetes: maximizing the applicability of clinical trials). None of the funders had any influence over the study design, analysis, or decision to submit for publication. The data contributors had no role in the funding, planning, conduct, or interpretation of this study.

**Competing interests:** I have read the journal's policy and the authors of this manuscript have the following competing interests: AC is part-funded by the National Institute for Health and Care Research (NIHR) Applied Research Collaboration Yorkshire & Humber, the NIHR Leeds Biomedical Research Centre, and Health Data Research UK, an initiative funded by UK Research and Innovation Councils, NIHR, and the UK devolved administrations and leading medical research charities. The views expressed in this publication are those of the authors and not necessarily those of the National Health Service, the NIHR, or the Department of Health and Social Care. NS declares grant funding from AstraZeneca, Boehringer Ingelheim, Novartis, and Roche Diagnostics; consulting fees from Abbott Laboratories, AbbVie, Amgen, AstraZeneca, Boehringer Ingelheim, Eli Lilly, Hanmi Pharmaceuticals, Janssen, Menarini-Ricerche, Novartis, Novo Nordisk, Pfizer, Roche Diagnostics, and Sanofi; payment for lectures or presentations from Abbott Laboratories, AbbVie, AstraZeneca, Boeringer Ingelheim, Eli Lilly, Janssen, Novo Nordisk, and Sanofi. All work was unrelated to this

higher risk populations, people with FI > 0.4 were generally absent. For SGLT2 inhibitors and GLP1 receptor agonists, there was a small attenuation in efficacy on HbA1c with increasing frailty (0.08%-point and 0.14%-point smaller reduction, respectively, per 0.1-point increase in FI), below the level of clinical significance. Findings for the effect of treatment on MACE (and whether this varied by frailty) had high uncertainty, with few events occurring in trial follow-up. A 0.1-point increase in the FI was associated with more all-cause adverse events regardless of treatment allocation (incidence rate ratio, IRR 1.44, 95% CI 1.35–1.54, *p* < 0.0001), adverse events judged to the possibly or probably related to treatment (1.36, 1.23, to 1.49, *p* < 0.0001), serious adverse events (2.09, 1.85, to 2.36, *p* < 0.0001), hypoglycaemia (1.21, 1.06, to 1.38, *p* = 0.012), baseline risk of MACE (hazard ratio 3.01, 2.48, to 3.67, *p* < 0.0001) and with withdrawal from the trial (odds ratio 1.41, 1.27, to 1.57, *p* < 0.0001). The main limitation was that the large cardiovascular outcome trials did not include data on functional status and so we were unable to assess frailty in these larger trials.

## Conclusions

Frailty was uncommon in these trials, and participants with a high degree of frailty were rarely included. Frailty is associated very modest attenuation of treatment efficacy for glycaemic outcomes and with greater incidence of both adverse events and MACE independent of treatment allocation. While these findings are compatible with calls to relax HbA1c-based targets in people living with frailty, they also highlight the need for inclusion of people living with frailty in trials. This would require changes to trial processes to facilitate the explicit assessment of frailty and support the participation of people living with frailty. Such changes are important as the absolute balance of risks and benefits remains uncertain among those with higher degrees of frailty, who are largely excluded from trials.

### Author summary

**Why was this study done?**

- Many people with type 2 diabetes are living with frailty, an age-related state of reduced physiological reserve.

- It is not known how the balance of risks and benefits of treatments for type 2 diabetes differs in the context of frailty.

- This study assessed frailty in clinical trials for 3 treatments for type 2 diabetes and assessed how common frailty was in these trials as well as whether the outcomes of trial participants were different depending on frailty.

**What did the researchers do and find?**

- It was possible to measure frailty in 34 trials for type 2 diabetes treatments, in which moderate frailty was present but people with the most severe degrees of frailty were not included in these trials.

- Trial participants with higher degrees of frailty had higher rates of cardiovascular events, adverse events (regardless of what treatment they were allocated to), and were less likely to complete the trial.

manuscript. MW declares grant funding from the National Institute for Health and Care Research, Medical Research Council and Biotechnology, and Biological Sciences Research Council. AA declares salary support from the National Institute for Health and Care Research Biomedical Research Centre (Oxford). DP declares consultancy fees from Bayer, AstraZeneca, and Bristol Myres Squibb. All work was unrelated to this manuscript. KR reports grants from Nova Scotia Health Research Fund, during the conduct of the study; personal fees from Ardea Outcomes , personal fees from Chinese Medical Association, personal fees from Wake Forest University Medical School Centre, personal fees from University of Nebraska - Omaha, personal fees from Australie New Zealand Society of Geriatric Medicine, personal fees from Atria Institute, personal fees from Fraser Health Authority, personal fees from McMster University, personal fees from EpiPharma Inc., outside the submitted work; in addition, Dr Rockwood has a patent Clinical Frailty Scale licensed to Enanta Pharmaceuticals, Inc., a patent Clinical Frailty Scale licensed to Synairgen Research Ltd, a patent Clinical Frailty Scale licensed to Faraday Pharmaceuticals, Inc., a patent Clinical Frailty Scale licensed to KCR S.A., a patent Clinical Frailty Scale licensed to Icosavax, Inc., a patent Pictorial Fit-Frail Scale licensed to Congenica, a patent Clinical Frailty Scale licensed to BioAge Labs Inc., a patent Clinical Frailty Scale licensed to Biotest AG, a patent Clinical Frailty Scale licensed to Qu Biologics Inc., a patent Clinical Frailty Scale licensed to AstraZeneca UK Limited, a patent Clinical Frailty Scale licensed to Cellcolabs AB, a patent Clinical Frailty Scale licensed to Pfizer Inc., a patent Clinical Frailty Scale licensed to W.L. Gore Associates Inc., a patent Clinical Frailty Scale pending to Cook Research Incorporated, a patent Clinical Frailty Scale pending to Rebibus Therapeutics Inc., and as part of Ardea Outcomes Inc. has a pending patent for Electronic Goal Attainment Scaling. Use of both the CFS and PFFS is free for education, research, and non-profit health care with completion of a permission agreement stipulating users will not change, charge for or commercialize the scales. For-profit entities (including pharma) pay a licensing fee, 15% of which is retained by the Dalhousie University Office of Commercialization and Innovation Engagement. After taxes, the remainder of the license fees is donated to the Dalhousie Medical Research Foundation. In addition to academic and hospital appointments, KR is co-founder of Ardea

What do these findings mean?

- At modest levels of frailty, the efficacy of type 2 diabetes treatment on blood glucose do not appear to differ; however, the baseline risk of both cardiovascular events (which treatments aim to prevent) and adverse events is higher in people with frailty.

- As people with severe frailty were generally excluded from these trials, there is greater uncertainty around risks and benefits for these people.

- The main limitation of this work is that it was not possible to measure frailty in the larger trials that assessed the impact of treatment on cardiovascular outcomes as these trials did not assess functional status. We argue that we need both the inclusion of people living with frailty, and more widespread reporting of the features required to assess frailty, to inform the treatment of people living with frailty.

## Background

Type 2 diabetes is an increasingly common condition associated with complications across multiple organ systems and reduced quality of life [1]. Type 2 diabetes becomes more prevalent as people age, with approximately half of all adults with type 2 diabetes aged over 65 years [2,3]. As a result, an increasing proportion of people with type 2 diabetes is living with frailty, an age-associated state of reduced physiological reserve [4]. Frailty is more common in people with type 2 diabetes than in similarly aged people without type 2 diabetes, affecting between 10% and 25% of people with the condition [5]. While the prevalence of frailty increases with age, it is also present in younger people (<65 years) with type 2 diabetes in whom it is also associated with adverse outcomes such as mortality, cardiovascular events, and hypoglycaemia [5,6]. In older people living with frailty, international guidelines recommend adjusting glycaemic targets [7]. However, these recommendations are typically based on observational data or extrapolating trial findings for older people in whom frailty has not been directly quantified [8,9]. Optimal treatments for people living with frailty, seeking to balance risks and benefits, often remain uncertain.

People with severe frailty are often explicitly excluded from randomized controlled trials [10–12]. Such exclusion makes uncertain whether trial findings apply to people living with frailty and type 2 diabetes. Frailty prevalence among participants is rarely quantified, as trials generally do not measure or report it. One approach to overcome this challenge is to apply the cumulative deficit frailty index (FI) retrospectively to trial data to estimate frailty among participants [10]. An FI is a count of age-related health deficits spanning multiple organ systems and functional domains [13]. This approach has been applied to individual trials including for hypertension, heart failure, and vaccination [14–17]. More recent studies have applied this approach across multiple trials [10,18,19]. However, these previous studies have neither systematically identified eligible trials nor have they synthesized estimates of heterogeneity in treatment efficacy across multiple trials.

This study aimed to identify frailty among participants of trials for newer glucose-lowering therapies for type 2 diabetes. By applying an FI to individual-level participant data, we aimed to assess (i) the prevalence and distribution of frailty across multiple trials, (ii) whether the efficacy of treatments varies depending on the degree of frailty, and (iii) the association between frailty and adverse events and whether individuals remain in the trial.

Outcomes (DGI Clinical until 2021), which in the past 3 years has had contracts with pharma and device manufacturers (Danone, Hollister, INmune, Novartis, Takeda) on individualized outcome measurement.

**Abbreviations:** DDP4, dipeptidyl peptidase 4; FI, frailty index; GLP1, glucagon-like peptide-1; GRADE, grading of recommendations, assessment, development, and evaluation; IPD, individual participant data; IRR, incidence rate ratio; MACE, major adverse cardiovascular event; MedDRA, Medical Dictionary of Regulatory Activities; OR, odds ratio; SGLT2, sodium-glucose cotransporter 2.

## Methods

### Identifying eligible randomized controlled trials

We included randomized controlled trials of three drug classes: sodium glucose cotransporter 2 (SGLT2) inhibitors, glucagon-like peptide-1 (GLP1) receptor analogues, and dipeptidyl peptidase-4 (DPP4) inhibitors. We first identified all potentially eligible trials through a systematic review before assessing the availability of individual participant data (IPD). The systematic review was conducted according to a pre-specified protocol as detailed elsewhere [20]. Briefly, two electronic databases (Medline and Embase) were searched from January 2002 to November 2022, supplemented by manual searching of trial registries. Trials were eligible for inclusion if they:

- Included adults (>18 years) with type 2 diabetes.

- Assessed the efficacy of SGLT2 inhibitors, GLP1 analogues, or DPP4 inhibitors, compared with either placebo or an active comparator (excluding within-class comparisons).

- Assessed HbA1c, major adverse cardiovascular events, or change in body weight as outcomes.

- Were registered phase-3 or phase-4 randomized controlled trials.

For each eligible trial, we explored the availability of IPD through the Vivli analysis platform. We then examined trial baseline data, metadata, and case report forms to identify variables that could be used to construct an FI. Trials were included in our analysis if they included data on a sufficient range of variables to allow valid construction of an FI (described in detail below). Ethical approval for IPD use was obtained from the University of Glasgow MVLS College Ethics Committee (Project: 200160070).

### FI construction

We assessed frailty using the cumulative deficit model to calculate an FI as a count of health deficits present within an individual divided by the number of possible deficits (in this case, the number of deficits measured within a given trial). The index ranges from 0 (no deficits present) to 1 (all possible deficits present) with higher numbers reflecting increased frailty. We selected deficits based on established criteria: they must be health-related; increase in prevalence with age; and be neither too rare (e.g., <1% in the target population) nor ubiquitous among older people (e.g., >80% prevalence by age 70) [21]. Deficits typically include long-term conditions, laboratory deficits, symptoms, and functional limitations. A valid FI should contain at least 30 deficits, spanning multiple domains and organ systems. The specific deficits included may vary between datasets, providing that their selection is based on the criteria described above.

Using these criteria, we constructed an FI for each trial by applying the standard approach to selecting deficits aligned with the baseline data of each trial. Deficits were selected from comorbidities (assessed from medical history data), laboratory and physical measurements (e.g., blood pressure), and patient-reported deficits (e.g., symptoms or functional limitations, assessed from baseline questionnaire data). To avoid an FI that was dominated by deficits from a single domain, we only included trials with data on comorbidities (both cardiovascular and non-cardiovascular), laboratory measures, and functional data. We excluded trials which did not collect each of these types of data. We excluded deficits with >10% missing data within a given trial.

### Outcomes

**FI distribution.** For each trial, we assessed the distribution of the FI. We also assessed the distribution of deficits in each of the following domains: cardio-metabolic comorbidities;

non-cardiometabolic comorbidities; laboratory deficits and physical measurements; and symptoms and functional limitations. In each case, the FI for each individual was calculated as the total number of deficits present divided by the total number of non-missing deficits.

**Efficacy.** For each trial, we assessed two outcomes: HbA1c and major adverse cardiovascular events (MACE). We assessed HbA1c as the follow-up value, adjusted for baseline HbA1c. End of follow-up was based on the primary endpoint of each respective trial. In trials including cross-over designs or an open label phase, we assessed efficacy prior to the cross-over period (which in each case was the primary endpoint). For our primary analysis, in individuals who were lost to follow-up prior to the primary endpoint the last recorded HbA1c value was carried forward. Where MACE was not a prespecified outcome in the trials, we identified MACE using Medical Dictionary of Regulatory Activities (MedDRA) codes applied to adverse event data. For the MACE analysis, participants were censored at the date of MACE, discontinuation of treatment, or end of trial follow-up (whichever happened earliest).

**Adverse events.** For each individual in each trial, we assessed the total number of adverse events (all-cause, regardless of their relationship with the study treatment), the number of adverse events assessed by the study investigators as being related to the study treatment (including events judged possibly, probably, or likely to be causally related to the study treatment), the total number of all-cause serious adverse events, and the number of hypoglycaemic events reported. For each of these outcomes, total follow-up time was also recorded based on the trial baseline, endpoint, and last available follow-up for participants who withdrew before the primary endpoint.

**Non-completion.** We assessed non-completion as any participant not completing the final trial visit for any reason (both intentional and unintentional withdrawal).

## Statistical analysis

All analyses were conducted on a secure analysis platform. This allowed analyses to be performed directly on the IPD but with export of summary data only, so that individuals cannot be identified. Our analysis therefore took the form of a two-stage IPD-meta-analysis approach, where sufficient statistics on covariates and model fits in each trial were extracted from the secure environment in the first stage, before being meta-analysed in the second stage [22].

Data on the distribution of the FI for each trial were summarized using statistical distributions. We fitted parametric cumulative distribution functions using the gamma, generalized gamma, log-normal, and Weibull distributions to FI distributions for each trial. Goodness-of-fit was assessed using the Kolmogorov–Smirnov test and by plotting observed FI distributions against each parametric cumulative distribution function. Parameters for the best-fitting cumulative distribution function models were exported to fully describe FI distributions. Based on recent guidance for constructing an FI [23], we did not define "frailty" based on any specific cut-off of the FI. Rather, we calculated the proportion of participants in each trial with FI values above a range of different thresholds (0.1, 0.2, 0.3, and 0.4). We also provide parameters for the best fitting distributions of each FI, allowing re-calculation of the proportion of participants above any given threshold of the FI.

To assess whether efficacy of treatment on HbA1c varied depending on the FI, we fitted a linear model with HbA1c as the outcome variable and the FI as the explanatory variable adjusted for baseline HbA1c, age, and sex, and including interactions between treatment allocation and each of FI, age and sex. We fitted a separate model in each trial directly on the IPD and exported the estimated coefficients and their variance-covariance matrix as sufficient statistics. We then meta-analysed these in a random-effects network meta-analysis using the *multinma* package to produce drug-class-level estimates for the frailty–treatment interaction,

adjusted for age and sex [24]. *Multinma* fits a separate intercept for each trial to ensure that randomization is preserved.

To assess consistency between direct and indirect estimates of the frailty × treatment interactions, we fitted further network meta-analysis restricting the dataset to trials in which the comparator arm was placebo. We then compared these (direct) estimated to the overall estimate (for which the comparator arms could include active treatment) from the primary analysis.

We used a similar approach for MACE in which we fitted a Cox proportional hazards model in each trial. For each trial, we fitted one model with a frailty–treatment interaction, and a further model in including frailty-, age-, and sex-treatment interactions. We excluded trials in which there were too few MACE overall to allow models to be reliably fit (typically < 15 events in total). We meta-analysed these models as described for HbA1c above to estimate drug-class level frailty–treatment interactions.

The association between the baseline FI and adverse events (total, serious, and hypoglycaemia) and trial withdrawal were assessed using negative binomial regression (estimating incidence rate ratios [IRR]) and logistic regression models (estimating odds ratios [OR]), respectively. Negative binomial models included an offset for follow-up time for each individual. We adjusted all models for age and sex. Coefficients for frailty (per 0.1-point increase) were exported from the analysis platform, along with their standard errors, and combined in a random-effects meta-analysis using the generic inverse variance method to estimate the IRR or OR of each outcome per 0.1-point increase in the FI. Finally, we used this same modelling approach to fit a separate model for each arm in each trial and combined these in a drug-class level network meta-analysis. This allowed us to estimate a frailty–treatment interaction for adverse events and attrition for each class (i.e., whether the association between any drug class and adverse events or trial non-completion was modified by the FI).

## Results

### Study selection and characteristics

We identified 34 trials of relevant drugs (SGLT2 inhibitors [$n = 10$], DPP4 inhibitors [$n = 15$], or GLP1 receptor analogies [$n = 10$]) for which we could obtain IPD and that collected data on a sufficient number and range of deficits to construct an FI.

The trial screening and inclusion process is summarized in Fig 1 and reported elsewhere [25]. Out of the 672 eligible trials, we were able to obtain IPD for 103 trials. Of these, most trials did not collect data on any patient-reported functional limitations (66/103, 64%) meaning that it was not possible to identify a sufficient range of deficits to assess the FI in these trials (as we judged assessment of functional status to be a prerequisite for our assessment of frailty among participants). A further 3 trials did assess functional deficits but did not include data on non-cardiometabolic comorbidities. These trials were also excluded from the analysis of frailty, as we were not able to assess comorbidities across multiple physiological systems. The trials for which we could assess frailty were similar to the wider body of eligible trials in terms of mean age, sex distribution, treatments assessed, and type of comparison (S1 Table). However, while we identified 23 trials that were designed and powered to assess cardiovascular outcomes and obtained IPD for 6 of these, none of these larger trials had sufficient data on function (5/6) or non-cardiometabolic comorbidities (1/6) for us to assess frailty. Change in HbA1c was the primary outcome in all of the studies included in this FI analysis.

Eligibility criteria for all trials are detailed in the supplementary appendix (S1 Appendix). Trials either reported no upper age limit (21/34) or excluded participants aged >80 (13/34). All trials included both male and female participants. All trials included participants with

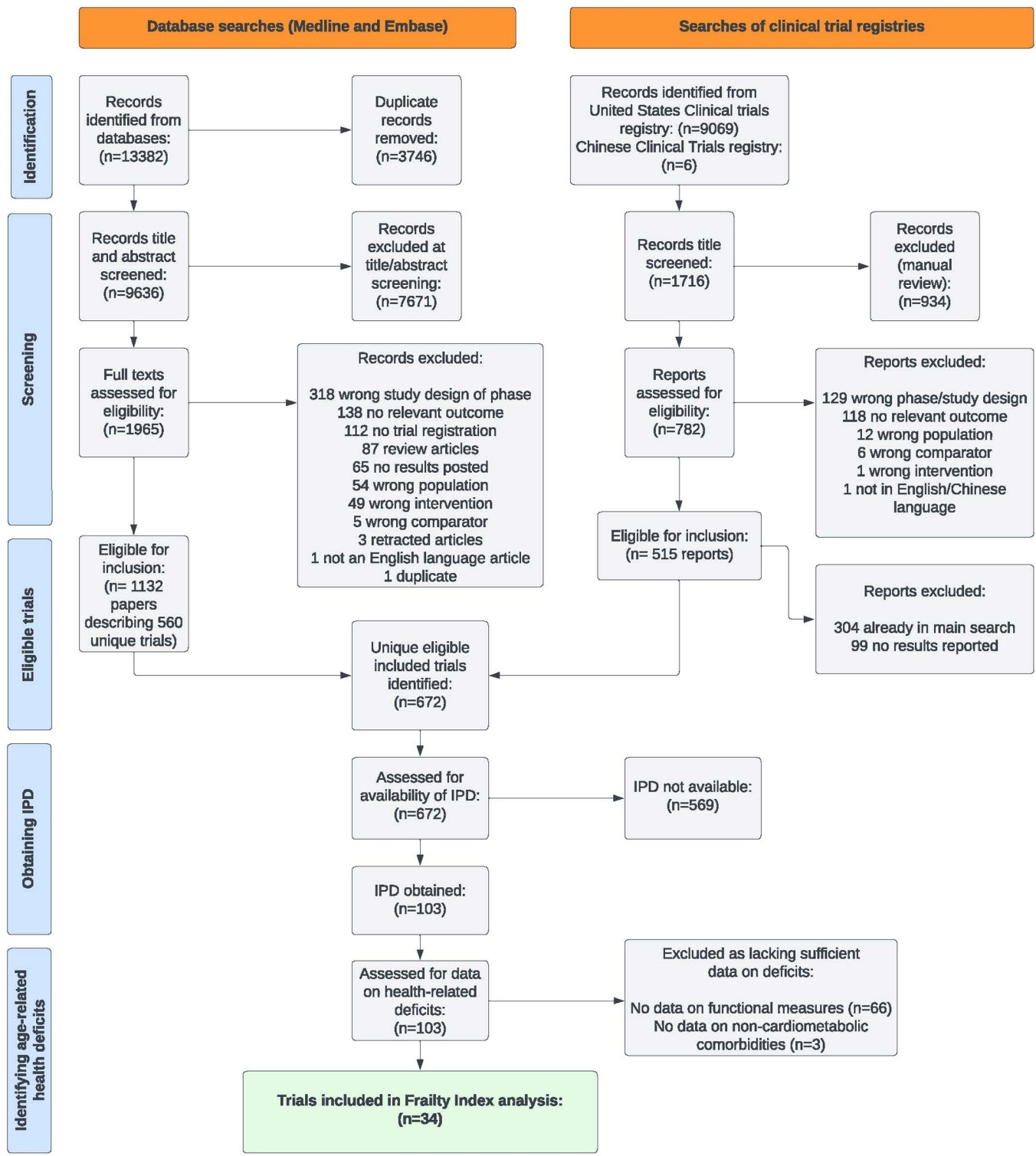

**Fig 1. Flow diagram of included trials.** This figure shows the identification and selection of included trials.

"inadequate glycaemic control", although the precise thresholds differed between the trials as did the background therapy on which this control was assessed. For 6/34 trials, participants could be drug naïve or on no current oral therapy; 18/34 trials included participants on the basis of being currently established on oral therapy with suboptimal glycaemic control; and 4/34 trials included participants who were already established on insulin therapy. The remaining 5/34 trials assessed specific sub-populations including older people (aged > 70, 2/34 trials) and people with kidney impairment (3/34 trials).

### Availability of deficits for the FI

The deficits included in the FI for each trial are summarized in Fig 2. The number of deficits within the FI in each trial ranged from 42 to 51. Comorbidities were consistently coded across all included trials using the MedDRA coding system. Most laboratory deficits were comparable across all trials. Symptoms and functional limitations were more variably quantified between trials, as different trials used different symptom and quality of life questionnaires.

### FI distribution

The distribution of the FI in each trial is shown in Fig 3 within categories indicating the target population of each trial. Frailty was rare in most trials. The proportion of trial participants for whom the FI was >0.2 ranged from 0.6% to 88.9% (median 9.5%, interquartile range 2.4% to 15.4%). Applying a higher cut-off of >0.3, the range was 0.004%–34.5% (median 0.5%, interquartile range 0.07%–1.5%). For 32/34 trials, less than 1% of trial participants had an FI >0.4 (S2 Table). The FI was higher, on average, in women compared with men, and in participants over 65 years (S3 and S4 Figs). When considering deficits from different domains separately, cardiovascular were more common compared with non-cardiometabolic comorbidities, laboratory deficits, and functional limitations (S5 Fig).

The mean age in these trials ranged from 53 to 75 years. However, even in trial participants aged >65 years, the prevalence of frailty was typically low (median 12.4% with FI > 0.2, interquartile range 4.6%–23.1%; median 0.7% with an FI > 0.3, interquartile range 0.1%–2.5%). In trials focused on older people (>70 years, 2 trials) or people with chronic renal impairment (3 trials, one of which focused on severe renal impairment), frailty prevalence was more variable with some trials showing a greater degree of frailty among trial participants (14.3%–88.9% with FI values above 0.2, 1.3%–34.5% with FI values above 0.3). In these trials, the upper limit of frailty (assessed by calculating the 99th percentile of the FI distribution) was between 0.31 and 0.48.

### Treatment efficacy

In all the included trials, change in HbA1c was the primary outcome. None of the included trials assessed MACE as a primary or secondary outcome (of the 6 MACE trials for which we had IPD, none collected sufficient data on function and/or comorbidity to allow the calculation of the FI). After identifying MACE within the adverse event data in the IPD, we were able to estimate MACE-treatment interactions in 12 of the 34 trials (models did not converge for the remining 22, which had <15 events each, precluding reliable estimation of covariate–treatment interactions). Findings from drug-level and drug class-level network meta-analyses of the interaction between FI and treatment efficacy are shown in Fig 4 (networks shown in S1 and S2 Figs).

In the main analysis for HbA1c, summarized in Fig 4, the pooled change in HbA1c with treatment was −1.2% (95% credible interval −1.4% to −1.0%), −1.2% (−1.6% to −1.1%), and −0.4% (−0.6% to −0.2%) for SGLT2 inhibitors, GLP1 receptor analogues, and DPP4

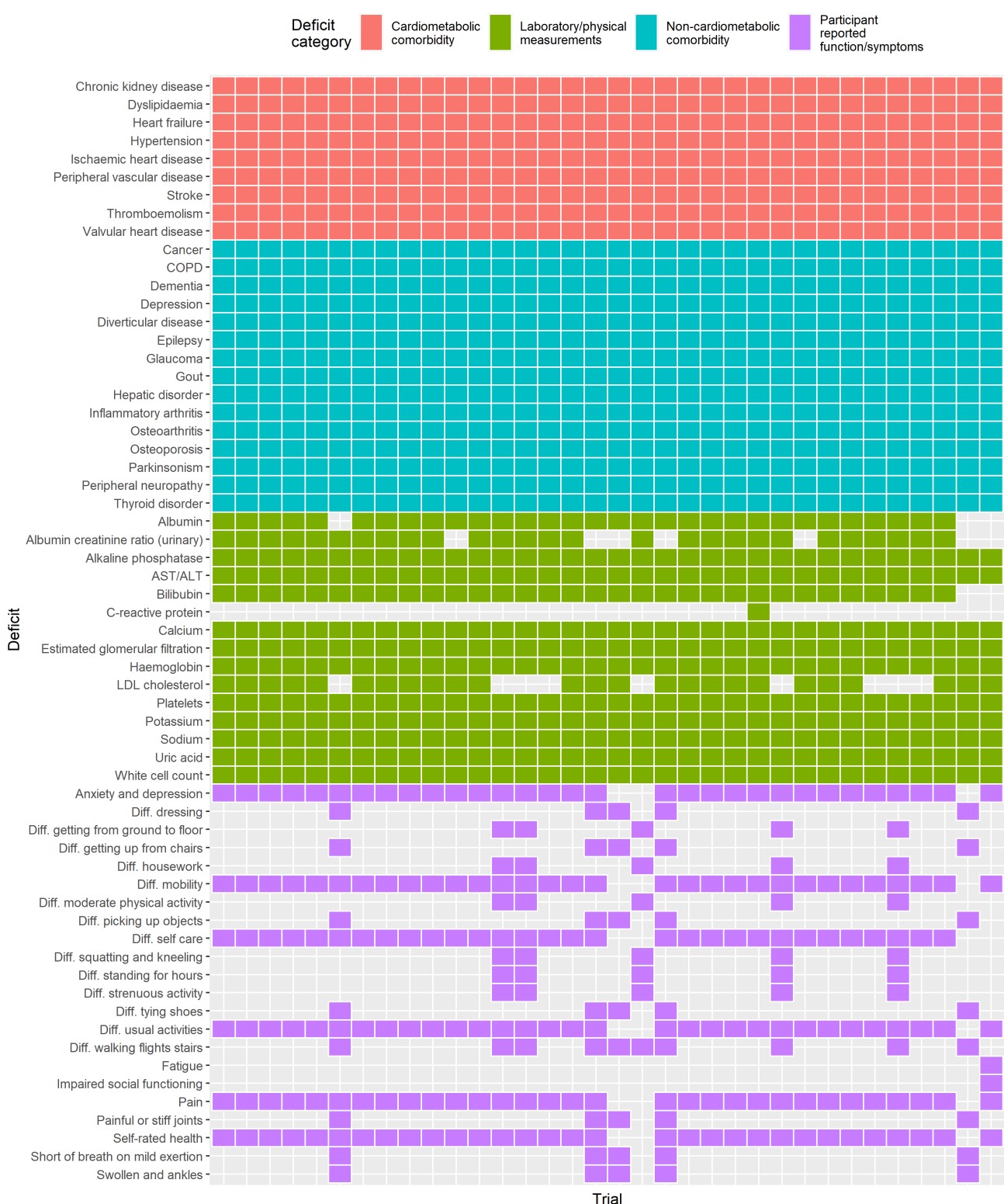

**Fig 2. Deficits included in the FI per trial.** This figure indicates, for each trial, what deficits were available within the trial IPD to be included within the FI. Colour is used to indicate the broad category of deficits (cardiometabolic comorbidities, non-cardiometabolic comorbidities, laboratory measures, and functional impairments). Each column indicates a single trial.

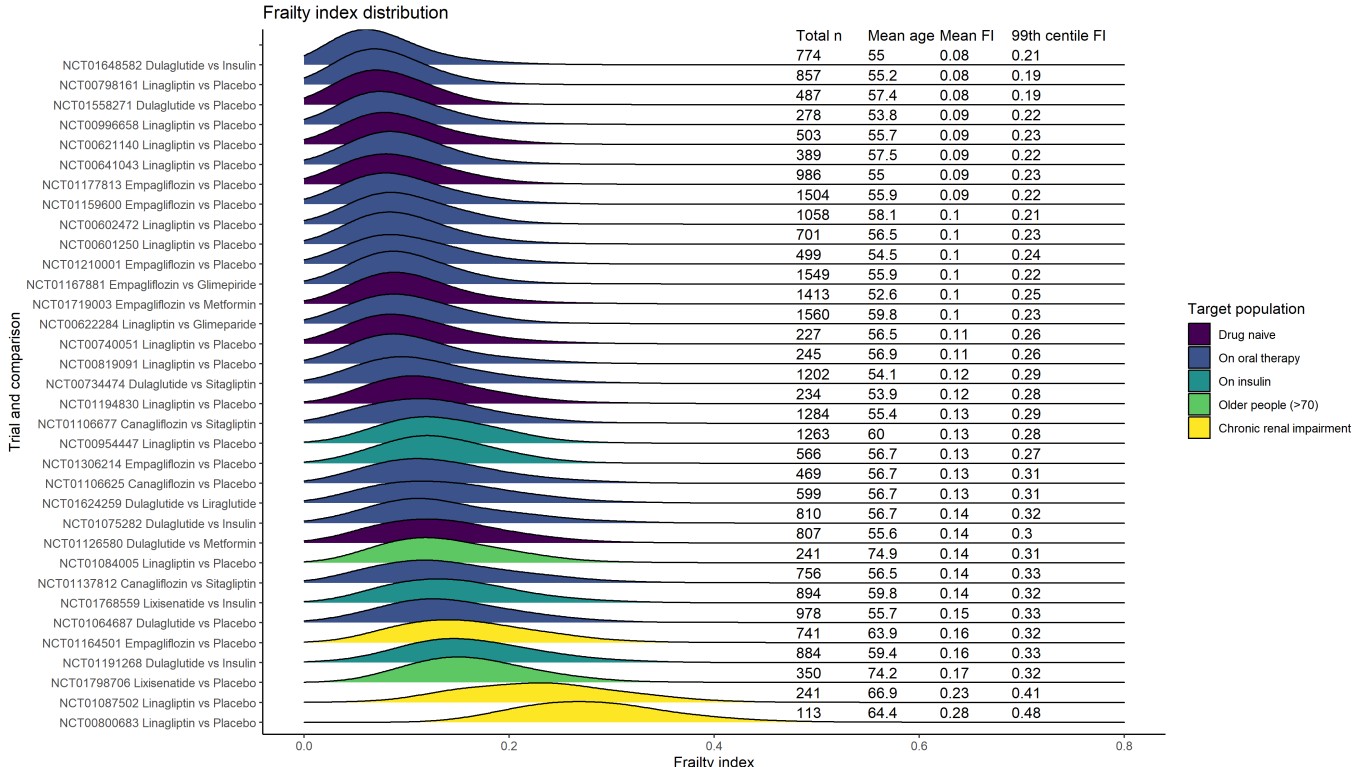

**Fig 3. Trial characteristics and FI distribution.** This figure shows the distribution of the FI among participants for each trial. All randomized participants are included. Colour indicates the target population of the trial based on the inclusion criteria. The y-axis indicates the treatment comparison and the trial registration. The 99th centile of the FI is reported as an upper limit of the FI within the trial population.

inhibitors, respectively, compared with placebo. For SGLT2 inhibitors and GLP1 receptor analogues, there was a small attenuation in the reduction in HbA1c with increasing frailty; however, the magnitude of this attenuation was small and below the threshold for clinical significance. In SGLT2 inhibitors, this reduction in HbA1c was slightly attenuated with increasing frailty (0.08% [0.02%–0.14%, $p = 0.029$] smaller reduction per 0.1-point increase in the FI). For GLP1 receptor analogues, there was also a small attenuation in the treatment effect (0.14% [0.04%–0.22%, $p = 0.019$] smaller reduction per 0.1-point increase in the FI). For DPP4 inhibitors, the frailty–treatment interaction included the null (0.04% [−0.01 to 0.10, $p = 0.23$]). These interaction terms were similar after limiting the analysis to trials the 23 trials with placebo comparisons (S6 Fig).

For MACE, the trials were small with few events and the estimates of overall efficacy were highly uncertain (Fig 4), limiting any inference about the association between frailty and efficacy. There was no statistically significant interaction between frailty and any treatment in terms of efficacy for MACE.

## Association between frailty and adverse events

The association between the FI and the overall incidence of adverse events (all-cause—regardless of perceived relationship with the study treatment), adverse events related to the study treatment (as judged by the trial investigators), serious adverse events, hypoglycaemic events, and trial non-completion are summarized in Fig 5. These associations do not take into account treatment allocation (i.e. they are expressing the association between the FI and the

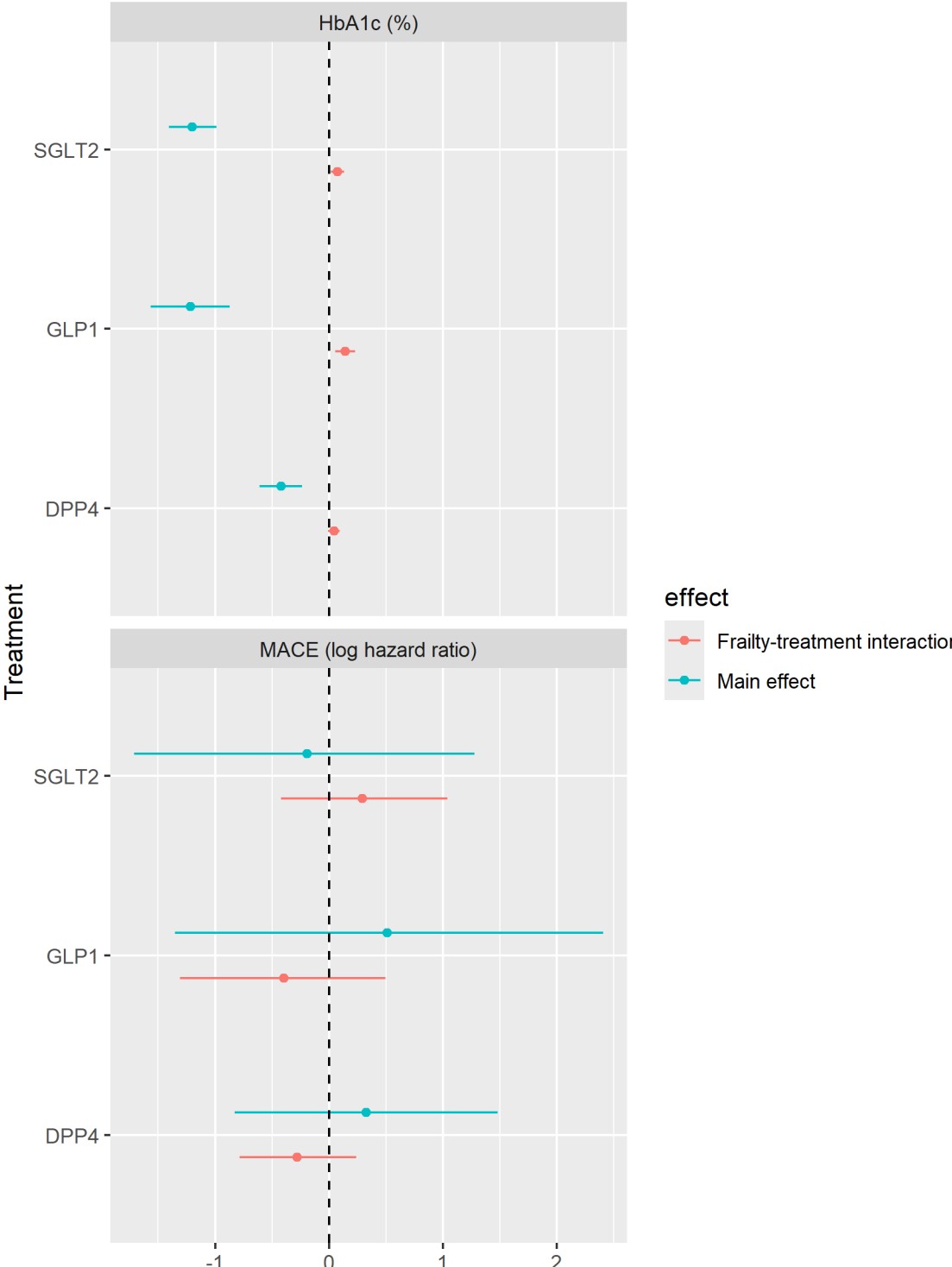

**Fig 4. Frailty and treatment efficacy.** This figure shows the results of the network meta-analysis for HbA1c (34 trials, top panel) and for MACE (11 trials, bottom panel). For each network, estimates for each class of drug are shown compared to placebo. Blue points indicate the estimated main treatment effect, with 95% credible intervals, in %-point reduction in HbA1c or log hazard ratio for HbA1c and MACE, respectively. These estimates indicate the treatment effect at FI = 0. Red points indicate the extent to which this treatment effect changes for each 0.1-point increase in the FI. These estimates are adjusted for age and sex.

## Association between frailty and adverse outcomes

| Outcome | N trials | Estimate (95% CI) | |
|---|---|---|---|
| Adverse events (all) | 34 | IRR 1.44 (1.35 to 1.54) | |
| Adverse events (treatment related) | 34 | IRR 1.36 (1.32 to 1.49) | |
| Serious adverse events | 34 | IRR 2.09 (1.85 to 2.36) | |
| Hypoglycaemia | 34 | IRR 1.20 (1.06 to 1.38) | |
| Trial non-completion | 34 | OR 1.41 (1.27 to 1.57) | |
| Major adverse cardiovascular event | 12 | HR 3.01 (2.69 to 3.37) | |

0.50    1.0    2.0    4.0
Estimate of association

**Fig 5. Association between FI and adverse events/trial attrition.** This figure shows the estimates association between baseline frailty and the incidence/odds of outcomes. Estimates are regardless of treatment allocation. Points indicate the estimate while bars show the 95% CI.

baseline rate of these events). In pooled analyses for all included trials, a 0.1-point increase in the FI was associated with an increased incidence of adverse events (IRR 1.44, 95% CI 1.35–1.54, $p < 0.0001$), treatment-related adverse events (1.36, 1.23–1.49, $p < 0.0001$), serious adverse events (2.09, 1.85–2.36, $p < 0.0001$), hypoglycaemia (1.20, 1.06–1.38, $p = 0.012$), and greater odds of non-completion (OR 1.41, 1.27–1.57, $p < 0.0001$). Higher FI was also associated with a greater hazard of MACE (12 trials, hazard ratio 3.01, 2.68–3.37, $p < 0.0001$).

When assessing whether the FI modified the association between any specific treatment and adverse events, there was no evidence that the association between any of the included classes of medication and adverse events varied by frailty status (i.e. there was no statistically significant frailty–treatment interaction when assessing adverse events or trials attrition).

## Discussion

In this analysis of IPD from 34 trials of newer glucose lowering treatments for type 2 diabetes, we found that, for most trials, frailty was rare among trial participants. Participants in trials focussed on older people or those with chronic renal impairment had a greater degree of frailty; however, even in these trials, severe frailty was uncommon and the upper limit of frailty was lower than is generally observed in unselected populations [26,27]. This low prevalence of frailty limits the inference that can be made about the efficacy and safety of these treatments in people living with frailty. We found a very modest attenuation of the efficacy of SGLT2 inhibitors and GLP1 receptor analogues on HbA1c with increasing frailty; however, this was below the threshold for clinical significance indicating that, at the modest levels of frailty observed in these trials, the efficacy of treatment on the surrogate endpoint of HbA1c was similar regardless of frailty. Assessment of the association between frailty and efficacy on cardiovascular outcomes was limited as the included trials had very few events and statistical uncertainty for this outcome was high. Frailty was associated with greater overall incidence of adverse events, serious adverse events, and hypoglycaemia and with greater odds of withdrawal before the end of the trial, indicating that people with a greater degree of frailty had

a higher baseline risk of these adverse outcomes. However, we found no evidence that frailty modified the associations between any specific trial treatment and adverse events.

These findings suggest that frailty is under-represented in trials for type 2 diabetes. Furthermore, the relatively low mean FI is some trials of older people (>70 years) in this analysis also demonstrates that simply recruiting older participants does not guarantee the inclusion of people living with frailty. This may be for several reasons. Trial exclusion criteria may exclude those living with frailty or those with limited life expectancy. Some physicians may be reluctant to put forward potential participants living with frailty when information on adverse events emerged early (e.g. genital infections with SGLT-2 inhibitors and nausea with GLP-1 receptor analogues). Furthermore, the demands of participating in trials, such as frequent clinic visits or frequent self-monitoring of blood glucose, may act bas barriers to the participation.

Judgements around treatment in people living with frailty therefore need to be based on careful balance between risks and benefits. In this context, our findings suggest that differences in efficacy for HbA1c, at least in those with a moderate degree of frailty, are small in magnitude and these judgements should therefore be based on the risk of adverse events and of competing risks [28]. We found that a higher FI was associated with a range of adverse health outcomes including hypoglycaemia and serious adverse events. This has important implications when considering the absolute risks and benefits of treatment. While we did not find any difference in the relative effect of treatments on adverse events by frailty status, where the baseline risk of adverse events is higher (as we found to be the case with frailty), then the absolute risk of any treatment associated with adverse events will be greater in these people. Frailty adds the additional dimension of greater vulnerability to decompensation, and it is possible that people living with more severe frailty may experience more severe consequences of adverse events. In this context, the exclusion of those with the most severe frailty from trials makes judging the absolute risks of treatment in this population challenging. It is also important to consider the potential benefits of treatment in absolute terms, and with respect to outcomes that are meaningful to patients. HbA1c is a surrogate marker which may precede clinical outcomes (of greater relevance to people with diabetes) by many years and may therefore be a lesser priority for some older people living with frailty. People living with frailty also had a higher risk of MACE. It is therefore possible that absolute benefits of treatment on MACE may be greater in people with higher FI. However, the net-benefit of treatment also depends on the balance between efficacy, adverse events, likelihood of treatment discontinuation, and competing risks such as non-cardiovascular death. The fact that cardiovascular deficits were the most common in these trials, and non-cardiovascular and functional deficits relatively rarer, may have influenced the magnitude of association with MACE, and may not necessarily reflect the balance of deficits seen in routine care [29]. Our capacity to assess for differences in efficacy for MACE was also limited by few events leading to lower power and high statistical uncertainty.

We need clinical evidence that is applicable to the populations who are being treated. Our findings demonstrate an important discrepancy between these trial populations (in which higher degrees of frailty were rare, even in trials focusing on older populations) and clinical practice (in which an increasing proportion of people with type 2 diabetes are living with frailty). We also show that it was only possible to assess frailty in a small subset of trials that assessed functional measures and not in the larger cardiovascular outcome trials which are arguably most influential in terms of clinical practice and guidelines. There is a need for trials including people with higher degrees of frailty, based on clinical outcomes which patients prioritize. For this to happen, there is a need for trials to adopt inclusion and exclusion criteria that enable people living with frailty to take part; to measure a broad range of comorbidities

and functional status to allow frailty to be reliably assessed (and ideally including additional domains, such as cognition, that were not assessed in these trials) [30]; to be designed to make it easy for older people living with frailty to participate (including allowing sufficient time for detailed baseline assessments and adopting approaches to outcome assessment that minimize the burden on participants); and to measure clinical endpoints that are relevant to people living with frailty rather than solely assessing surrogate markers [31].

Previous meta-analyses of treatments for type 2 diabetes have demonstrated that SGLT2 inhibitors and GLP1 receptor analogues reduce the risk of all-cause mortality, MACE, and of end-stage kidney disease with high certainty of evidence according to the grading of recommendations, assessment, development, and evaluation (GRADE) framework [32]. We did not apply GRADE to this analysis as our aim was not to assess the overall efficacy of these agents (which is well established, and for which our estimate would be very uncertain due to the exclusion of the large, high-quality cardiovascular outcome trials that have established cardiovascular benefits). Our analysis highlights an important limitation of this evidence base for people living with frailty, which is that frailty can only be assessed in a small subset of trials. Cardiovascular outcome trials typically recruit "higher risk" populations based on cardiovascular disease or risk factors. It is currently not clear if these higher-risk trial populations also have a greater degree of frailty. Our findings that trials focusing on older populations or people with chronic kidney disease had higher degrees of frailty but excluded those with the most severe degrees of frailty highlights that the adequate representation of people with frailty cannot be assumed even when inclusion criteria target individuals with a greater health burden.

Other studies have applied the FI to IPD for single trials (including trials for heart failure, hypertension, and vaccination studies). These studies found no significant difference in treatment efficacy across the spectrum of frailty included [14,15,33,34]. Analyses of single trials are typically under-powered to detect differences in treatment efficacy by participant characteristics. Two previous studies have applied the FI to multiple trials; however, these did not assess treatment efficacy [10,19]. By combining 34 trials, our IPD network meta-analysis provides considerably greater statistical power to assess differences in treatment efficacy across individual-level characteristics than previous analyses and found a small but statistically significant difference in efficacy on glycaemic outcomes by frailty. However, these findings should be interpreted in light of the generally low prevalence of frailty in most of the included trials and it is still possible that the balance of efficacy and safety may differ in people living with more advanced frailty who are generally excluded from these trials.

Strengths of this study include the systematic identification of trials and the inclusion of IPD from a large number of trials. However, our analysis remains limited by the fact that IPD is not available for all eligible trials (103/672, 15%) and among those that we accessed, only 33% (34/103) collected sufficient data to calculate the FI. Notably, we were unable to assess frailty in the larger trials for which MACE was the primary outcome. This highlights the considerable challenge to analysing frailty across multiple trials, both in terms of data availability and in identifying a sufficient range of deficits within trial data. Our FI was constructed according to established methods for selecting and analysing deficits. Comorbidities and laboratory deficits were consistently recorded across the included trials. However, there was a limited number and range of functional deficits. This limits the granularity of the FI based on trial data and supports recent calls for standardizing the collection of functional data within trial participants [30]. While the FI is designed to allow flexible application across datasets with different variables, it remains possible that these differences in included deficits could explain some of the differences between trials. The included trials also lacked any measures in some domains, such as cognition. While the ability of the FI to predict adverse outcomes is robust to the selection of deficits, provided they cover a range of domains and fulfil the

required criteria, omitting deficits from a domain entirely may have a greater impact on the FI [35]. Finally, the FI is one of several measures of frailty, and we were not able to assess others (e.g. the frailty phenotype [36]) due to a lack of relevant data within these trials.

In conclusion, our findings show that frailty is under-represented in these trials with people living with the most advanced frailty largely excluded. We found a clinically negligible reduction in efficacy for glycaemic outcomes of newer glucose-lowering treatments in people living with frailty; however, in general, the low prevalence of frailty in these trials limited the inference about efficacy in the context of frailty. Frailty is associated with greater baseline risk of adverse events and with premature withdrawal from trials, but not with observable difference in the safety of specific treatments. Faced with these uncertainties, it is likely that with greater degrees of frailty, the risk of adverse outcomes may increase further, and the time to accrue benefits from treatment is likely to be less. Decisions of whether to initiate treatment in people living with frailty should carefully reflect individual goals and priorities, consider guideline recommendations to relax HbA1c-based treatment targets for people living with frailty, and be cognisant of the limited randomized evidence for the balance of risks and benefits in people living with frailty.

## Supporting information

**S1 Appendix.  Detail of links to systematic review protocol, full results of data extraction and risk of bias assessment for all trials, pre-specification of FI analyses, and definitions for all FI deficits.**
(DOCX)

**S2 Appendix.  Analysis code for all results presented.**
(DOCX)

**S1 Table.  Comparison of all eligible trials, trials for which IPD were available, and trials for which the FI could be constructed.**
(DOCX)

**S2 Table.  Percentage of participants with FI values at a range of thresholds.**
(DOCX)

**S1 Fig.  Network diagram for all trials included in the analysis of HbA1c.**
(PNG)

**S2 Fig.  Network diagram for all trials included in the analysis of MACE.**
(PNG)

**S3 Fig.  FI distributions for participants older and younger than 65 years for each trial.**
(PDF)

**S4 Fig.  FI distributions for male and female participants for each trial.**
(PDF)

**S5 Fig.  Distributions of deficits within each domain of the FI.**
(PNG)

**S6 Fig.  Frailty × treatment interactions for HbA1c based on the main analysis compared to an analysis of direct comparisons only (restricted to trials versus placebo).**
(PNG)

**S1 Data.  Underlying data exported from secure analysis platform.**
(ZIP)

**S1 Checklist. PRISMA Checklist.**
(DOCX)

## Acknowledgments

This manuscript is based on research using data from data contributors Lilly, Boehringer Ingelheim, Sanofi and Johnson, and Johnson that has been made available through Vivli, Vivli, has not contributed to or approved, and is not in any way responsible for, the contents of this publication. This study was carried out under YODA project 2022–5124 and used data obtained from the Yale University Open Data Access Project, which has an agreement with Janssen Research and Development, LLC. The interpretation and reporting of the research data are solely the responsibility of the authors and do not necessarily represent the official views of the Yale University Open Data Access Project or Janssen Research and Development, LLC.

## Author contributions

**Conceptualization:** David A. McAllister, Peter Hanlon.

**Data curation:** Heather Wightman, Elaine Butterly, Lili Wei, Ryan McChrystal, Peter Hanlon.

**Formal analysis:** David A. McAllister, Peter Hanlon.

**Funding acquisition:** Peter Hanlon.

**Investigation:** Elaine Butterly, Ryan McChrystal, David A. McAllister, Peter Hanlon.

**Methodology:** David Phillippo, Sofia Dias, Nicky Welton, Andrew Clegg, Miles Witham, Kenneth Rockwood, David A. McAllister, Peter Hanlon.

**Project administration:** Peter Hanlon.

**Supervision:** Peter Hanlon.

**Visualization:** Peter Hanlon.

**Writing – original draft:** Heather Wightman, Peter Hanlon.

**Writing – review & editing:** Elaine Butterly, Lili Wei, Ryan McChrystal, Naveed Sattar, Amanda Adler, David Phillippo, Sofia Dias, Nicky Welton, Andrew Clegg, Miles Witham, Kenneth Rockwood, David A. McAllister.

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
