## [Editor Report · Decision Letter 0]

17 Sep 2024

Dear Dr Hanlon,

Thank you for submitting your manuscript entitled "Frailty in randomised controlled trials of glucose-lowering therapies for type 2 diabetes: An individual participant data meta-analysis of frailty prevalence, treatment efficacy, and adverse events" for consideration by PLOS Medicine.

Your manuscript has now been evaluated by the PLOS Medicine editorial staff as well as by an academic editor with relevant expertise and I am writing to let you know that we would like to send your submission out for external peer review.

Please re-submit your manuscript within two working days, i.e. by Sep 19 2024 11:59PM.

Feel free to email me at pdodd@plos.org or the team at plosmedicine@plos.org if you have any queries relating to your submission.

Kind regards,

Pippa

Philippa C. Dodd, MBBS MRCP PhD

Senior Editor

PLOS Medicine

pdodd@plos.org

---

## [Decision Letter · Decision Letter 1]

16 Dec 2024

Dear Dr Hanlon,

Many thanks for submitting your manuscript "Frailty in randomised controlled trials of glucose-lowering therapies for type 2 diabetes: An individual participant data meta-analysis of frailty prevalence, treatment efficacy, and adverse events" (PMEDICINE-D-24-03055R1) to PLOS Medicine. The paper has been reviewed by four subject experts and a statistician; their comments are included below and can also be accessed here: [LINK]

As you will see, the reviewers somewhat mixed in their opinions of the study given the paucity of data for patients with frailty that were included in the original trials. After substantial discussion, the editors agreed strongly with Reviewer 1 that the study is important for drawing attention to the evidence-practice gap in this population and that the data are valuable to report from that standpoint, even if definitive conclusions cannot be drawn from the data. There were also some questions raised about the methodology and statistical approach. After discussing the paper with the editorial team and an academic editor with relevant expertise, I'm pleased to invite you to revise the paper in response to the reviewers' comments. We plan to send the revised paper to some or all of the original reviewers, and we cannot provide any guarantees at this stage regarding publication.

Given the upcoming holidays, we ask that you submit your revision by Wed, Jan 8th. However, if this deadline is not feasible, please contact me by email, and we can discuss a suitable alternative. Please also feel free to contact directly with any questions (hvanepps@plos.org).

Kind regards,

Heather

Heather Van Epps, PhD

Executive Editor

[on behalf of]

Philippa Dodd, MBBS MRCP PhD

Associate Editor

PLOS Medicine

pdodd@plos.org

Comments from the reviewers:

Reviewer #1:

On careful reading of the paper I had no major methodological comments. I note that this is a first revision (I was not involved in the previous round of review). [Editors’ note: this reviewer appears to be referring to the ‘R1’ in the manuscript number; the editors will clarify that this was the first round of review, and there is no need to respond to this comment.]

The authors have described a very important evidence-practice gap. Increasingly our hospitals and primary care health centres are filled with older people living with frailty and yet, we have few reliable data to guide our treatment decisions. Historically, this was often due to clinical trials having upper age limited related to retirement age (e.g. 65 years), or due to erroneous reading of cholesterol epidemiology (with the historical age limit of 75 years). Despite recommendations that upper age limits should only be used if there was good evidence to suggest recruiting older people is definitely harmful, it remains surprising that these diabetic trials still had age limits.

I would recommend that the authors also include in the methods the original denominator of potential trials (as seen in Figure n = 596). Even with the 6% of trials with IPD available, less than half could contribute, and of those, remarkable few people living with frailty were recruited. The key message from this paper is that the frail have not been appropriately recruited into diabetes trials, and we therefore have substantial uncertainty of the benefits of many new treatments. Given that most people with diabetes are older people, and many have frailty (and corresponding co-morbidity, with the resulting polypharmacy) there is additional uncertainty whether the addition of these newer agents will have net harm or benefit.

I think the authors have missed an opportunity for a call to arms: the drug regulators and the government funders should particularly request data on frail older people, and new practical arrangements must be supported to allow the recruitment of those living with frailty. The authors quite rightly point out that one certain aspect of their paper is that the event rates are driven up with increasing frailty, so event driven trials will achieve outcomes quite quickly in this age group. The pharmaceutical industry is terrified that their nice new drug will be associated with unfortunate adverse events if used in those with frailty and even if not actively avoided, the actual practicalities of trial procedures will often lead to only the robust being able to manage intense trial monitoring of follow-up procedures. Older people with frailty can be recruited but you probably need more time for baseline assessment, you need to measure important aspects of health such as cognition and function, you need frailty friendly trial procedures (such as home visits or remote assessment) and you need to ensure the procedures are not too onerous. Trials within cohorts, and the use of routinely collected data (as used, for example in the RECOVERY trials in Covid) provide examples of some work arounds. Frailty can also be electronically calculated with many electronic patient records.

Overall, this is an important paper that illustrates a major evidence practice gap in the care of older people with frailty (a very large proportion of many hospitals worldwide). Richard Lindley. University of Sydney

Reviewer #2 (statistical review):

Firstly, I would like to thank the authors for their recent submission. This submission was an IPD analysis of frailties impacts on treatment, including adverse events. Overall, the work is well written and presented. I have some points below for your consideration:

1. Abstract: mean age is reported as a range. Is this the mean age across the included trials? Please clarify in text.

2. For the linear models (e.g. HbA1c), adjustments were made for baseline, age and sex. Did the authors consider use participant ID as a random factor in a mixed effect model? Or was this deemed not necessary due to the data type (e.g. no repeated measurements)? It may be worthwhile conducting a mixed model and comparing model fits between the current model and mixed model.

3. How was consistency between the direct and indirect evidence assessed?

4. Heterogeneity is mentioned in the discussion, stating there was increased statistical power to assess heterogeneity. I could have missed the details in the text/supplementary material. I struggled to find evidence of this being reported. Could the authors please clarify in text what the heterogeneity assessment statistics were and the results obtained.

5. I would recommend the inclusion of GRADE for NMAs, see: https://www.bmj.com/content/381/bmj-2022-074495

6. There is no specific guidance for IPD NMAs, but this could aid in the interpretation of the results.

Reviewer #3:

Wightman et al., leveraging individual participant data from over 30 randomized trials, examine glucose-lowering therapies and the relevance of frailty (defined via deficit accumulation) in adults with type 2 diabetes. In general, this is an impressive study that tries to fill some of the current data gap underlying clinical guidelines (i.e. recommending higher glycemic targets in adults with frailty, multimorbidity, or otherwise limited life expectancy).

The major point of this work is that the full spectrum of frailty is generally not well-represented in type 2 diabetes trials of newer agents. While in some sense I think this is a result that everyone knows, or at least suspects, the value is really in showing a precise quantification of just how limited the inclusion of frail older adults has been in these trials. I, for one, expected the frailty prevalence estimate to be low, but not generally well below 10%. It is also then not surprising that there was little evidence of treatment effect heterogeneity in these trials, given the more limited range of deficit accumulation scores that tended to be represented.

The major limitation of the work is acknowledged by the authors. They were unable to include the cardiovascular outcome trials due to data limitations in constructing a valid frailty index, which limits what can be said about the benefit side of the equation in terms of cardiovascular prevention. This also makes me wonder how representative the subset of trials examined is of the broader set of trials in this population, given that over 50 of the available trials needed to be excluded.

One limitation that the authors should discuss is the focus on treatment effect heterogeneity on the relative scale for MACE and adverse events. This is certainly the typical approach and is also more statistically tractable from a meta-analysis perspective. However, heterogeneity on the absolute scale is likely the more relevant metric for clinical decision making (i.e. a 20% reduction in frail patients is likely a much larger absolute difference versus a 20% reduction in robust patients with lower event rates). The authors could cite the PATH statements in Annals of Internal Medicine, which have a relative concise discussion of this issue.

Minor Comments:

1. Line 41. Recognizing that most thresholds for frailty indices are in some sense ad-hoc, the choice of >0.24 for frailty seems a bit weird. I've seen >0.20, >0.21, or even >0.25 used, but never >0.24. Was this a data-driven choice or based on some other justification?

2. Line 50-51. I find the results around MACE adverse events confusing. How is this different from the trial MACE outcomes for which the author's report a limited number of events and a high degree of uncertainty? The abstract reads as if the authors have not much to say about MACE, but then somehow there are more statistically certain results?

3. Line 118-119 and 386-387. While the deficit accumulation approach is mostly robust to the precise composition of included deficits, provided they span appropriate domains with sufficient density, there has been some work showing that the impact of variable deficit composition is generally modest, though still not inconsequential (see Shi et al., PMID 32274807). I worry about it a little more in this context given the wide heterogeneity with respect to the functional domain, though I'd agree with the authors that I don't this would strongly impact the study's results.

4. Lines 143-144. I'm curious about the choice to censor at the point of treatment discontinuation. That seems inconsistent with an intention-to-treat approach. Certainly other estimands are justifiable, the authors should just be explicit in terms of what they are estimating, along the lines of the definitions in work such as Kahan et al. (PMID 36790803).

5. Lines 161-163. Do the authors have a citation for this statement? While I think this is generally true, it does depend on the model somewhat. For some models, it's more of an approximately correct statement, and I can definitely come up with model structures (i.e. any sort of shared covariance across trials) where this result would not hold.

6. Line 171-177. Did the analyses of HbA1c outcome for length of follow-up at all? I see that in the later analyses adverse events, but doesn't seem to be accounted for with HbA1c?

7. Lines 202-209. Can the authors provide any other high-level descriptions that might speak to how the excluded trials might be different from the included trials? Obviously there is the situation of the cardiovascular longer-term outcome trials, but are there any other relevant differences? Perhaps differences along the lines of age, sex, race/ethnicity?

8. Lines 398-399. While I think the data here is supportive of this statement, in some sense, the guideline recommendation is a bit of a different animal. Most of these trials reflect the situation of initiating treatment, which is not the same as treatment changes or deprescribing after continued long-term use. There is a distinction there that needs to be made clear.

Reviewer #4:

Overall, a well designed and constructed topical study. Frailty and Diabetes have a reciprocal relationship and this make the study even more topical. Important subject at a time where we are finding greater absolute benefits in frail older adults with diabetes given SGLT2i and GLP- 1 RAs i terms of reducing MACE.

I cannot fault the written account of the way in which the literature searching was undertaken to identify the selected studies. The diagrams and figure are excellent. However, my concerns are two fold:

1. These were retrospective studies (with all the accompanying shortfalls of such) in which frailty was clearly not measured and was not part of the original design or research question: constructing the FI from deficits that are available in the trials information is, in my view, subject to bias and potentially inaccurate. In many ways, I view this as cutting corners in frailty detection and identification.

2. Frailty detection even using the FI was low in this study but I am not surprised considering the very restricted age range of subjects in this study - you need to go beyond 70 years to increase the likelihood of identifying frailty and complications such as chronic renal disease which increase substantially the risk of frailty.

My overall impression is that this could have been a very important study for clinical guidelines development if the participants had been mainly in their 70s and 80s, and that accepted frailty measures had been recorded by the individual study investigators at the start of their respective studies. However, I accept that finding such studies may be difficult at the present time. Unfortunately, it means that the findings in this study do not take us forward.

Reviewer #5:

The authors reported on a meta-analysis of RCTs of interactions between frailty index (deficit accumulation model) and antidiabetic drugs such as SGLT2 inhibitors and GLP-1 receptor agonists for HbA1c improvement, MACE, adverse events, and drop-out. The study design and results are very interesting, but concerns have been raised regarding the selection and assessment of the frailty index and its components.

1. In the original study by Rockwood et al, the Frailty Index included chronic conditions, functional status (IADL and ADL), and geriatric syndromes, but not laboratory findings. Because the Frailty Index included laboratory findings, would it have been more likely to produce adverse events? Would similar results be obtained using a frailty index that excluded laboratory findings?

2. The study defined frailty as a Frailty Index of 0.24 or higher. The cut-off appears to be too high, which would have reduced the prevalence of frailty?

3. Did the ethics of the study population influence the results?

---

* Please upload any figures associated with your paper as individual TIF or EPS files with 300dpi resolution at resubmission; please read our figure guidelines for more information on our requirements: http://journals.plos.org/plosmedicine/s/figures. While revising your submission, please upload your figure files to the PACE digital diagnostic tool, https://pacev2.apexcovantage.com/. PACE helps ensure that figures meet PLOS requirements. To use PACE, you must first register as a user. Then, login and navigate to the UPLOAD tab, where you will find detailed instructions on how to use the tool. If you encounter any issues or have any questions when using PACE, please email us at PLOSMedicine@plos.org.

* Please ensure that the study is reported according to the PRISMA-IPD (https://www.equator-network.org/reporting-guidelines/prisma-ipd/) guideline and include the completed PRISMA-IPD checklist as Supporting Information. When completing the checklist, please use section and paragraph numbers, rather than page numbers. Please add the following statement, or similar, to the Methods: "This study is reported as per PRISMA-IPD guideline (S1 Checklist)."

FIGURES AND TABLES

SUPPLEMENTARY MATERIAL

REFERENCES

SYSTEMATIC REVIEWS & META-ANALYSES

* As above, please report your SR/MA according to the PRISMA-IPD guidelines provided at the EQUATOR site. https://www.equator-network.org/reporting-guidelines/prisma-ipd/ Please provide the completed PRISMA checklist as Supporting Information. When completing the checklist, please use section and paragraph numbers, rather than page numbers. Please add the following statement, or similar, to the Methods: "This study is reported as per the Preferred Reporting Items for Systematic Reviews and Meta-Analyses (PRISMA)-IPD guideline (S1 Checklist)."

* Abstract: Please report your abstract according to PRISMA for abstracts (https://doi.org/10.1371/journal.pmed.1001419) following the PLOS Medicine abstract structure (Background, Methods and Findings, Conclusions). Please ensure you provide dates of search, data sources, number of studies included, types of study designs included, eligibility criteria, and synthesis/appraisal methods.

---

## [Decision Letter · Decision Letter 2]

28 Jan 2025

Dear Dr. Hanlon,

Thank you very much for re-submitting your revised manuscript "Frailty in randomised controlled trials of glucose-lowering therapies for type 2 diabetes: An individual participant data meta-analysis of frailty prevalence, treatment efficacy, and adverse events" (PMEDICINE-D-24-03055R2) for review by PLOS Medicine.

I have discussed the paper with my colleagues and the academic editor and it was also seen again by two of the original reviewers. I am pleased to say that provided the remaining editorial and production issues are dealt with we are planning to accept the paper for publication in the journal.

[LINK]

We ask that you submit your revised manuscript by February 4th. Please email me directly if this deadline is not feasible or you have any questions or concerns (hvanepps@plos.org). We look forward to receiving your revised manuscript.

Kind regards,

Heather

Heather Van Epps, PhD

Executive Editor

PLOS Medicine

hvanepps@plos.org

Requests from Editors:

1. Abstract. Please include a sentence describing the main limitation(s) of the study at the end of the Methods & findings section.

Comments from Reviewers:

Reviewer #2:

I would like to thank the authors for their careful consideration of the points raised by the reviewers.

Reviewer #3:

I thank the authors for a thorough and robust response to my original critiques. I have nothing further of substance to add.

[LINK]

---

## [Editor Report · Decision Letter 3]

27 Feb 2025

Dear Peter,

Sorry for the delay in being able to return this manuscript to you to make the revisions you have outlined. It requires some technical steps on our end once the post editorial accept stage has been reinitiated.

You can use this link to submit your revised manuscript when ready:

[LINK]

As outlined in your email, please include an outline for how the coding error occurred and what the impact is for the conclusions. We will need to discuss this as an editorial team and may need to consult with the peer reviewers as well.

We appreciate the transparency in highlighting the need for these changes, and we will endeavour to provide you with an updated decision as soon as possible when we receive the revised manuscript.

We look forward to receiving the revised manuscript by Mar 06 2025 11:59PM.   

Sincerely,

Becs

Rebecca Kirk, PhD, MBA (Pronouns: she/her)

Associate Editorial Director

Remote, UK

rkirk@plos.org

PLOS | A catalyst for better

Public Library of Science

UK Registered Office Address: Nine Hills Road, Cambridge, CB2 1GE, UK

Company registered in California, USA with UK Establishment Office in England and Wales

California (US) corporation #C2354500

UK Company # FC031758

---

## [Decision Letter · Decision Letter 4]

11 Mar 2025

Dear Dr Hanlon, 

On behalf of my colleagues and the Academic Editor, Sanjay Basu, I am pleased to inform you that we have agreed to publish your manuscript "Frailty in randomised controlled trials of glucose-lowering therapies for type 2 diabetes: An individual participant data meta-analysis of frailty prevalence, treatment efficacy, and adverse events" (PMEDICINE-D-24-03055R4) in PLOS Medicine.

As you know, before your manuscript can be formally accepted you will need to complete some formatting changes, which you will receive in a follow up email. Please be aware that it may take several days for you to receive this email; during this time no action is required by you. Once you have received these formatting requests, please note that your manuscript will not be scheduled for publication until you have made the required changes.

PRESS

Sincerely, 

Rebecca Kirk 

Senior Editor 

PLOS Medicine